

# Citizen science enhances understanding of sea turtle distribution in the Gulf of California

Stephanie J. Rousso[1,2], María Dinorah Herrero Perezrul[2], Agnese Mancini[3], Alan A. Zavala-Norzagaray[3,4] and Jesse F. Senko[5,6]

[1] Upwell Turtles, Monterey, CA, United States of America
[2] Centro Interdisciplinario de Ciencias Marinas, Instituto Politécnico Nacional, La Paz, BCS, Mexico
[3] Grupo Tortuguero de Las Californias A.C., La Paz, BCS, Mexico
[4] Centro Interdisciplinario de Investigación para el Desarrollo Integral Regional Unidad Sinaloa, Instituto Politécnico Nacional, Guasave, Sinaloa, Mexico
[5] School of Ocean Futures, Arizona State University, Tempe, AZ, United States of America
[6] School for the Future of Innovation in Society, Arizona State University, Tempe, AZ, United States of America

Corresponding authors
Stephanie J. Rousso,
ssmith0129@alumno.ipn.mx
Jesse F. Senko, jesse.senko@asu.edu,
jesse.senko@gmail.com

## ABSTRACT

Citizen science is a valuable tool for addressing spatial distribution gaps in endangered species, especially in data-limited regions. Given the logistical and financial challenges of studying migratory species, this cost-effective approach contributes to strategic conservation planning. The Bay of La Paz, located in Baja California Sur, México, is considered an ecologically important region within the larger Gulf of California. Due to its rich biodiversity and abundant natural resources, the region serves as a hub for ecotourism and fishing, affording diverse opportunities for community-based scientific initiatives. This paper examines the value of citizen science contributions from three diverse community groups (*i.e.*, coastal residents, SCUBA divers, and artisanal fishers) to help scientists obtain information on sea turtle distribution in the Bay of La Paz. Our findings represent the first records of loggerhead turtles (Caretta caretta) in the bay based on ten citizen science reports that include live and dead turtles (either observed swimming, as bycatch, or stranded), providing new information on an endangered pelagic species identified in a coastal bay. Although the sample size is small, our diversity of sources and sighting types highlight the value of collaborative citizen science initiatives in complementing traditional research methods.

## INTRODUCTION

Citizen science is a cost-effective method that engages local community groups through public outreach (*Bonney et al., 2016*). Also known as participatory research or community-based science, this approach can guide scientific research by shaping research questions, improving project design, and enabling new discoveries (*Conrad & Hilchey, 2011*). Scientific advancements utilizing citizen science are now evolving into legal frameworks

and national scientific policies. For example, the U.S. enacted the federal Crowdsourcing and Citizen Science Act (5 U.S. Code §3724), allowing federal agencies to integrate citizen science into key mission areas.

Citizen science initiatives can help document the distribution and identify new habitats for endangered, threatened, and protected (ETP) marine species, such as sea turtles. Engaging the public in an active role to share sightings of sea turtles enhances data collection and fosters a greater understanding and support for marine conservation (*Chandler et al., 2017*). For example, in Southern California, the National Oceanic and Atmospheric Administration (NOAA) invites the public to report sea turtle sightings *via* the West Coast Sea Turtle Sightings form on their website (https://www.fisheries.noaa.gov/west-coast/science-data/green-turtle-research-and-conservation-southern-california). The NOAA Citizen Science Strategy provides the framework for scientists to leverage crowdsourced data, significantly enhancing our understanding of sea turtle distribution (*Volz et al., 2021*). A study utilizing eight years of citizen science data from the San Gabriel River, California demonstrated the effectiveness of this approach in monitoring sea turtle populations in accessible coastal areas (*Massey et al., 2023*). Similarly, NOAA encouraged SCUBA divers in La Jolla Bay, California, to submit photographs of underwater sea turtle sightings which yielded results indicating fidelity and residency duration at specific dive sites (*Hanna et al., 2021*).

Given that sea turtles are highly migratory and often traverse international waters, engaging fishers in México as citizen scientists to report sea turtle sightings can play a crucial role in identifying spatial distribution gaps. This approach has been successfully demonstrated in fisheries to improve stock sustainability (*Fulton et al., 2019*), and can also lead to better-informed cross-border conservation strategies for sea turtles (*Peckham & Maldonado-Díaz, 2012*; *Senko, Jenkins & Peckham, 2017*). In the North Pacific, juvenile loggerhead turtles are known to spend up to 20 years foraging off the Pacific Coast of México's Baja California peninsula after hatching on nesting beaches in Japan (*Briscoe et al., 2016*; *Briscoe et al., 2021*; *Seminoff et al., 2014*; *Peckham et al., 2011*). However, until recently, the Baja California Peninsula was thought to act as a land barrier to loggerhead movement into the Gulf of California (GOC) (*Seminoff et al., 2004*). Citizen science data from fishers reporting accidental captures of sea turtles off the coast of Sinaloa, México revealed the first record of loggerheads in the GOC (*Zavala-Norzagaray et al., 2017*), enabling researchers to study post-release movements (*Sandoval-Lugo et al., 2020*). Analysis of the tracked loggerheads identified five distinct foraging sites closely linked to regions of high productivity based on chlorophyll-a and sea surface temperature: the Upper GOC, Bay of Los Angeles (BLA), Guaymas Basin, Tiburón Island, and the Central GOC (Fig. 1). However, none of the tracked individuals were observed in the Bay of La Paz (BLP), one of the most ecologically and socioeconomically important bays in the GOC. Here, we present the first record of loggerhead turtles in the BLP based on opportunistic citizen science sightings submitted from three diverse community groups: coastal residents, artisanal fishers, and SCUBA divers. These records contribute to recent studies regarding the spatial and temporal distribution of loggerhead turtles in the North Pacific, supporting
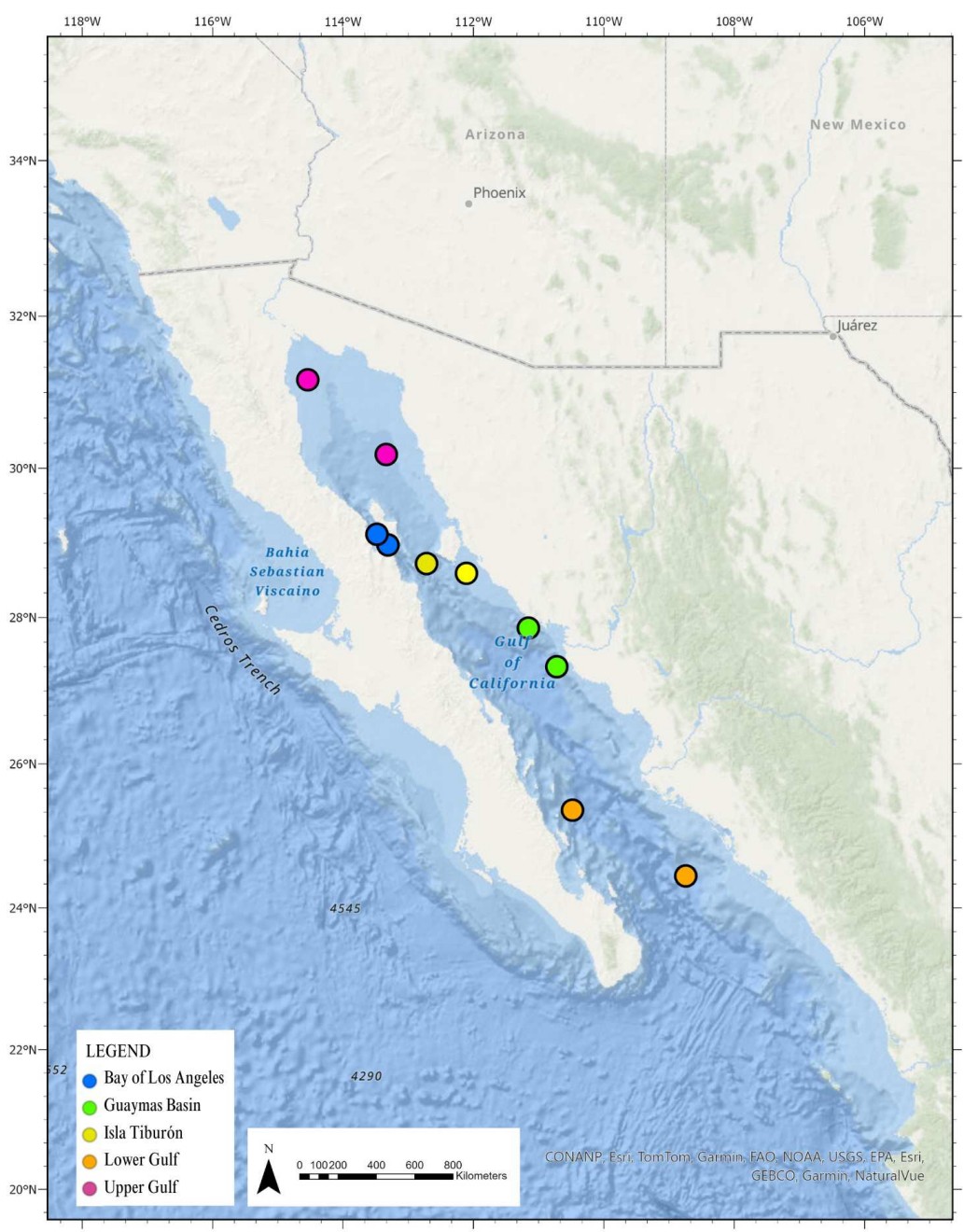

**Figure 1  Regional view of Northwest Mexico highlighting the five foraging areas published by** *Sandoval-Lugo et al. (2020)*, **with permission to recreate.** Colored circles represent different foraging areas, north and south boundaries as published by *Sandoval-Lugo et al. (2020)*, with permission to recreate.

broader international conservation efforts of this endangered population (*IUCN, 2022*) and underscoring the importance of diverse, collaborative citizen science initiatives.

## MATERIALS & METHODS

### Site description

The BLP is located on the eastern shoreline of the Baja California Peninsula and influenced by the oceanographic connection and economic importance to the GOC (*Cisneros-Mata, 2010*). Since the BLP is open to the GOC, we used the quadrilateral delineated bathymetry described by *del Monte-Luna et al. (2005)* as a general guide to select citizen science sightings of loggerheads reported in the BLP. The coastal shallows are characterized with extensive seagrass beds and mangrove forests, nurturing a wide array of marine species, (*Félix-Pico et al., 2006*) while the San Cosme Corridor and San Lorenzo Channel facilitate water exchange with the open GOC, shaping the mixture of diurnal and semidiurnal currents and habitats (*Guevara-Guillén et al., 2018*). The BLP is characterized by a shallow extensive continental shelf and is connected to the GOC by two passages: the Northern Alfonso basin and the San Lorenzo Channel (*Torres-Hernández et al., 2022*). Underwater canyons create refuge zones and nutrient-rich upwelling areas, contributing to the bay's rich biodiversity and habitat diversity that attract all five Eastern Pacific sea turtle species (*Johnson et al., 2016*). The BLP is a mixed-use region with marine protected areas, coastal residential and ecotourism areas, and artisanal fishing camps (Fig. 2).

### Citizen science species validation

Since 2016, Upwell Turtles, a NGO based in California, US, in conjunction with the Mexican Sea Turtle Network, Grupo Tortuguero de Las Californias (GTC) have received hundreds of sea turtle sightings reported opportunistically by various coastal community sectors. Selecting only sightings referencing loggerhead turtles within the general boundaries of the BLP, ten citizen science reports were considered ''research-grade'' based on the iNaturalist.org criteria (*iNaturalist, 2022*). iNaturalist.org is a citizen science platform that considers research-grade sightings when at least two researchers can confirm the species. Reports that contained clear photographs were included if two researchers from Upwell or GTC verified the species identification of loggerhead. In lieu of photos, sightings were included if Upwell or GTC verified the sea turtle in the field. If only the carapace of the turtle was available, in the case of stranded dead turtles, we used NOAA diagrams to verify the number of scutes and the shape characteristic to confirm the sighting as a loggerhead.

### Citizen science report classification

Each of the turtle sightings located within the BLP verified as loggerhead species were given an unique identification number (01-10). Sightings were grouped into three categories: (1) community sector (coastal residents, recreational divers, or small-scale fishers); (2) sea turtle status (live or dead) and; (3) type of report (stranded, bycatch, swimming), presented in Table 1. Sightings were mapped out using ArcGIS however, since the precision of citizen scientist-provided coordinates varied, the location of sightings was determined using available georeferencing details and cross-referenced with satellite imagery in Google
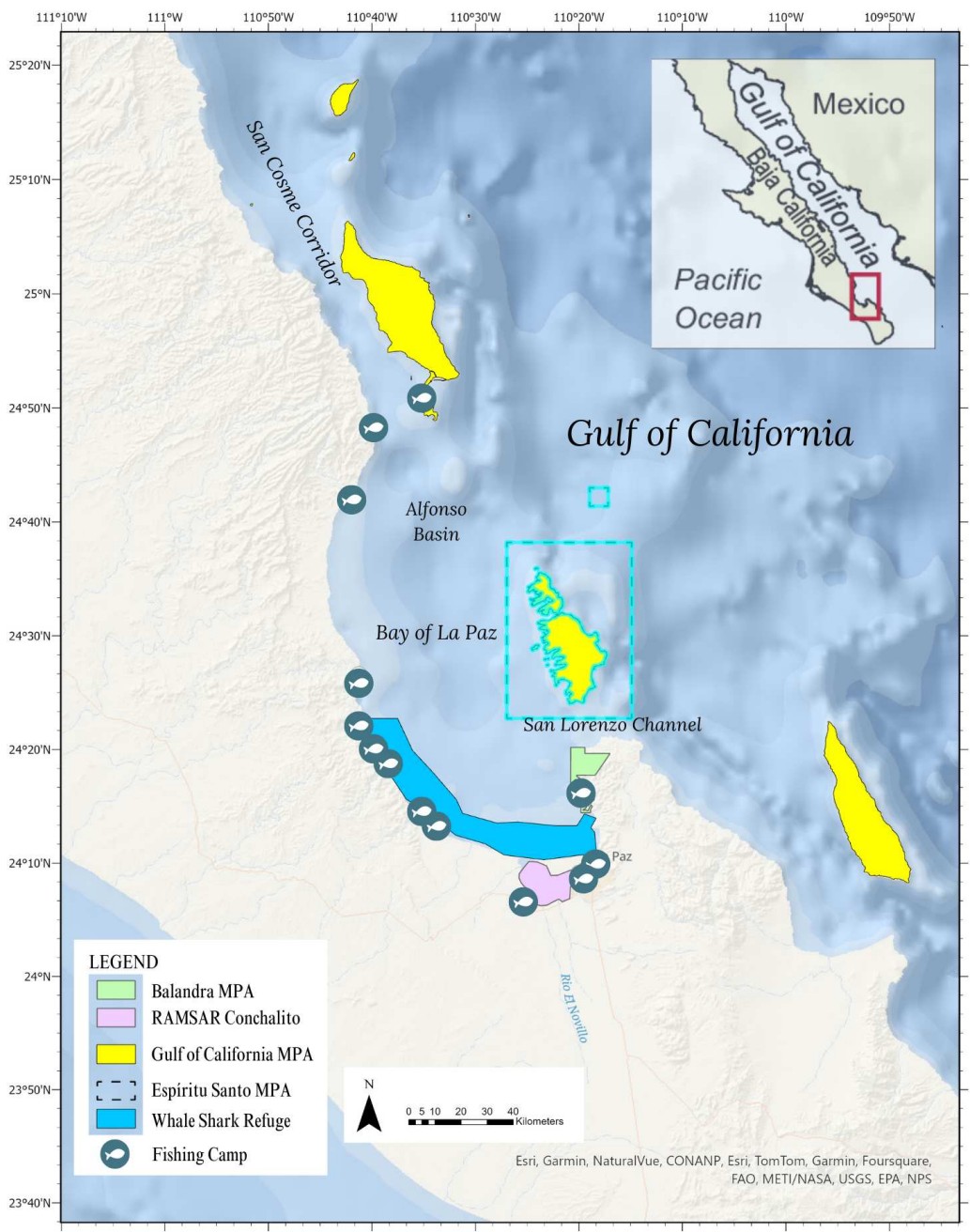

**Figure 2** **Study area of the Bay of La Paz, Baja Sur, México showing the marine protected areas at various levels from National, RAMSAR, and UN Heritage.** A portion of the extensive chain of protected islands throughout the Gulf of California is highlighted that pertains to the Bay of La Paz.

Earth. Locations of fish camps and polygons of marine protected areas (MPA) were added to the GIS maps for reference (Fig. 2).

**Table 1  Citizen science reported sightings of loggerhead turtles in the Bay of La Paz, México.**

| ID | Site name | Date | Latitude | Longitude | Status | Sighting | Figure | Origin | Verified | Photos |
|----|-----------|------|----------|-----------|--------|----------|--------|--------|----------|--------|
| 01 | CICIMAR | Jun 2018 | 24.14145 | −110.35197 | Dead | Stranded | 5.a | Resident | Field | Yes |
| 02 | MOGOTE | Jul 2018 | 24.18886 | −110.48245 | Dead | Stranded | 4.a | Resident | Photo | Yes |
| 03 | EL BAJO | Jul 2018 | 24.6146 | −110.3652 | Live | Swimming | 6.b | Diver | Video | Yes |
| 04 | EL CAJETE | Aug 2018 | 24.25768 | −110.54869 | Dead | Bycatch | N/A | Fisher | Field | No |
| 05 | LA DUNA | Jan 2019 | 24.247937 | −110.59781 | Dead | Stranded | 5.b | Resident | Field | Yes |
| 06 | SAN JUAN de la COSTA | Feb 2019 | 24.342694 | −110.66918 | Dead | Stranded | N/A | Resident | Field | Yes |
| 07 | MALECON | Dec 2019 | 24.171625 | −110.30731 | Dead | Stranded | N/A | Resident | Photo | Yes |
| 08 | MOGOTE 2 | Nov 2020 | 24.180556 | −110.44083 | Dead | Stranded | 4.b | Resident | Photo | Yes |
| 09 | FANG MING | Jul 2022 | 24.4303 | −110.3744 | Live | Swimming | 6.a | Diver | Video | Yes |
| 10 | LA BASTILLA | Sept 2022 | 24.30387 | −110.5243 | Dead | Bycatch | N/A | Fisher | Field | Yes |

## RESULTS

The ten confirmed loggerhead turtle sightings observed here present the first record of loggerheads in BLP based on citizen science. Sightings were reported from three diverse community groups: coastal residents, SCUBA divers, and one artisanal fisher. The sightings selected were reported between 2017 and 2023 and all were georeferenced within the BLP (Fig. 3). Among these sightings, 20% were documented as live sightings, while the remaining 80% were recorded as dead sightings (Table 1).

Within the collection of ten loggerhead sightings in the BLP, we include seven photographs which show one of the bycatch turtles (Fig. 4), four of the six stranded turtles (Figs. 5–8) and the two live swimming turtles reported by SCUBA divers (Figs. 9 and 10). According to one fisher, two separate dead loggerhead turtles were reported as bycatch over a span of four years (Fig. 4). There were six reports of stranded turtles submitted by various individual coastal residents, both local and international seasonal residents. Stranded turtles were reported along the western and southwestern portion of the BLP in the same vicinity of the bycaught turtles adjacent to several fishing communities. The cause of mortality of the stranded turtles could not be confirmed without a full necropsy and due to various stages of decomposition. Yet, two of the stranded turtles were photographed by citizen scientists with anthropogenic debris (Figs. 5 and 6). By contrast, there were two live turtle sightings that were both reported as swimming by individual recreational SCUBA divers on two different occasions at two distinct dive sites (Figs. 9 and 10). Both of the live sightings occurred during recreational dives within the Islas del Golfo de California MPA which comprises all the islands throughout the GOC.

## DISCUSSION

The sightings reviewed in this paper represent the first records of loggerhead turtles in the BLP based on citizen science data. The sightings, contributed by multiple community groups—coastal residents, SCUBA divers, and artisanal fishers—include live and dead turtles, categorized into three types: swimming, bycatch, and stranded. This diversity

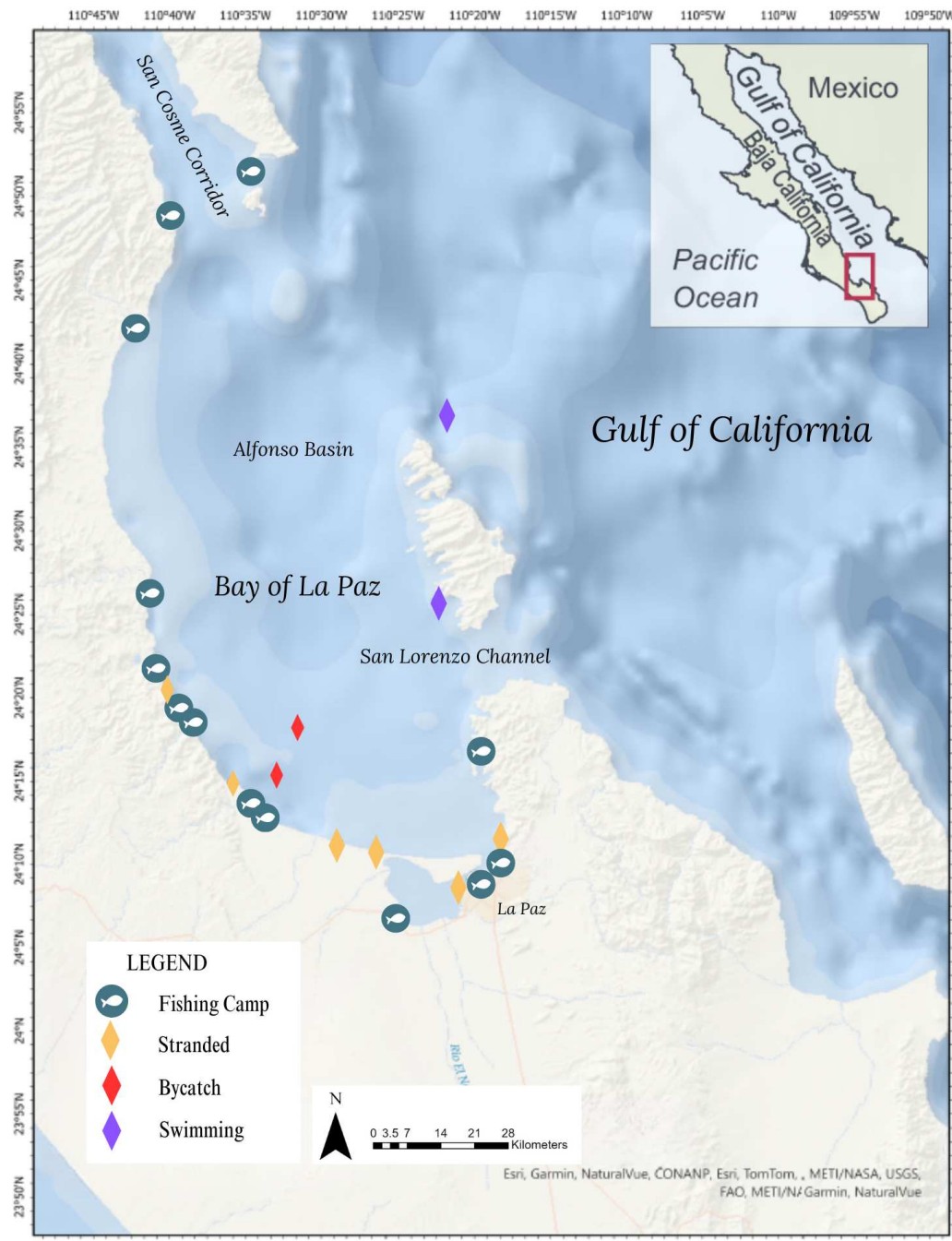

**Figure 3  Spatial distribution of loggerhead turtle sightings reported through citizen science in the Bay of La Paz.** Fish icons represent small fishing camps. Colored diamonds represent citizen science reports.

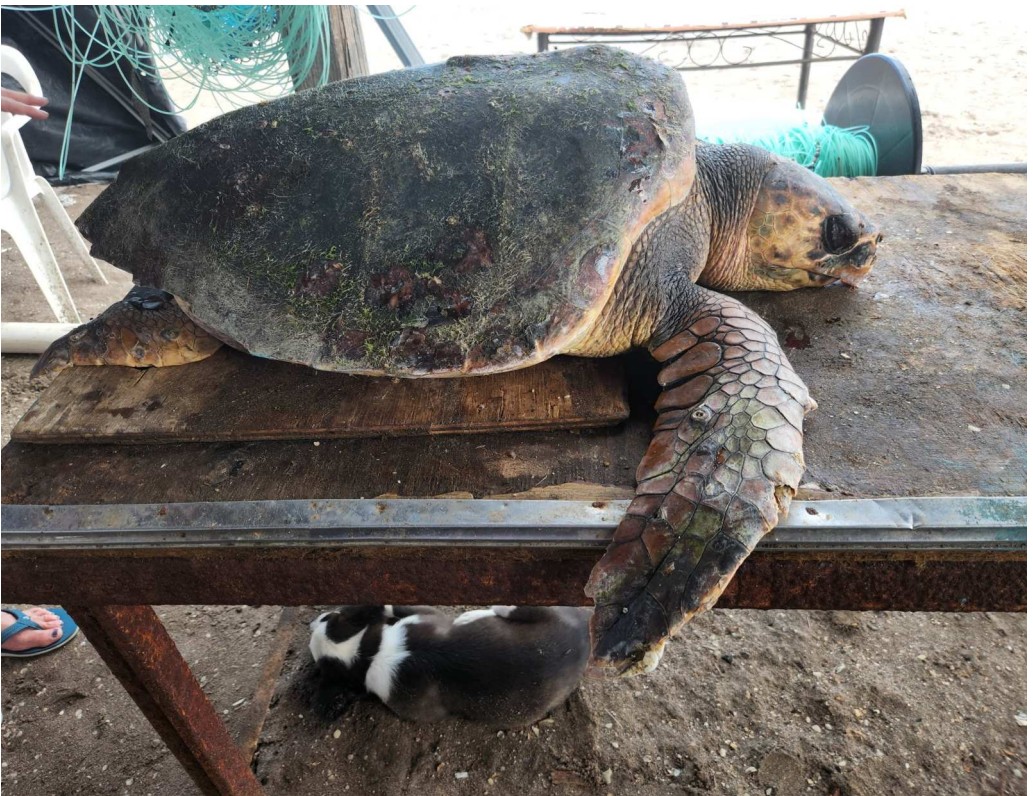

**Figure 4** **Loggerhead turtle reported verbally by a fisherman as bycatch.** ID 10. Photo credit: Zoe Diacou.

of sources and sighting types underscores the value of citizen science in providing new information on marine species distribution, particularly in under-studied and ecologically important regions such as the BLP.

While we cannot stipulate the cause of the strandings, it should be noted that citizen science provided valuable information on a pelagic species identified in a coastal bay. In the North Pacific, loggerhead turtles are typically considered pelagic, especially during their juvenile life stage, with the subpopulation in this region known to nest in Japan and forage off the Pacific coast of the Baja California Peninsula (*Briscoe et al., 2016*; *Seminoff et al., 2014*; *Peckham et al., 2011*). The presence of loggerheads in a coastal bay raises questions as to the significance of the bay and the GOC for foraging and habitat use of this endangered subpopulation. Our findings complement the recent findings of the first record of loggerheads off the coast of Sinaloa (*Zavala-Norzagaray et al., 2017*). Future research efforts should track loggerhead turtle movements in the BLP to better understand how and the extent to which they are using the bay.

The two live sightings were documented by different divers on separate occasions at two island dive sites within the Islas del Golfo MPA. These divers have also reported sightings of other sea turtle species at these and other dive sites. Given that the BLP is a popular destination for SCUBA divers and snorkelers, these findings present an opportunity to

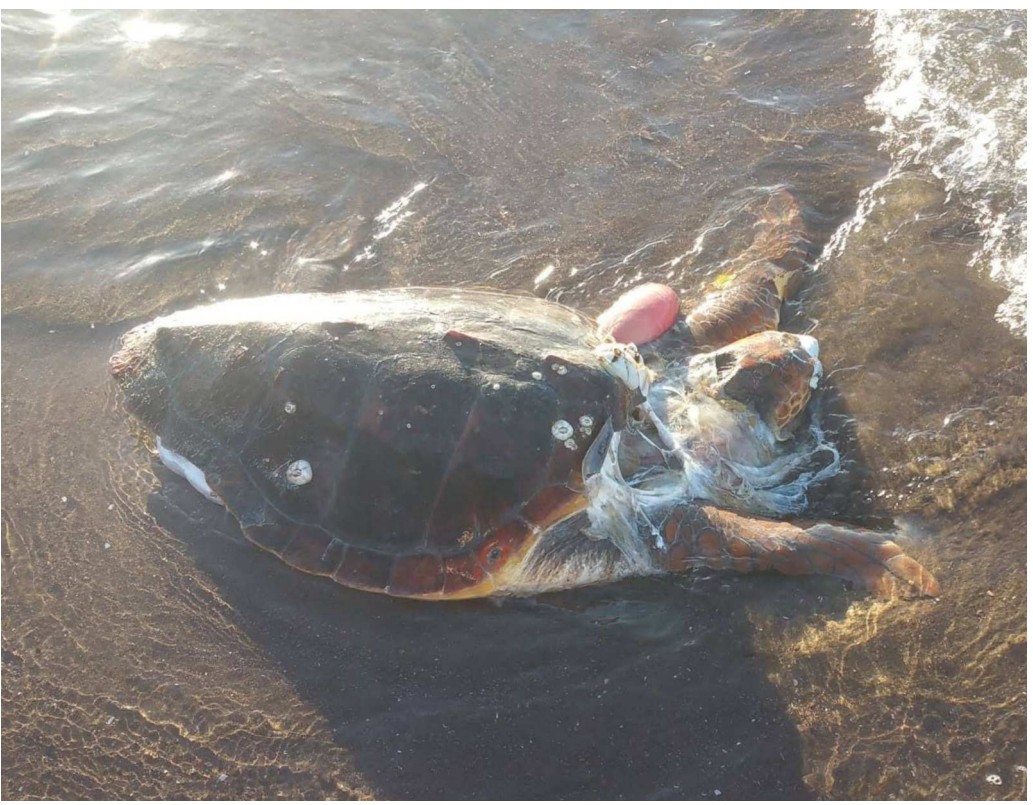

**Figure 5** **Stranded loggerhead turtle reported via social media with evidence of plastic debris.** ID 02. Photo Credit: Guri Sejzer.

co-create a structured citizen science program that could generate a robust sample size for scientific analysis. SCUBA divers are a reliable subset for citizen science as they frequently revisit the same sites, providing insights into site fidelity, sighting frequency over time, and seasonal usage of specific dive sites (*Hanna et al., 2021*). Implementing a standardized protocol for volunteer SCUBA divers to capture photographs can help identify individual turtles and offer valuable information on site fidelity and habitat use, similar to the approach used in La Jolla, California (*Hanna et al., 2021*). This analysis could enhance our understanding of the effectiveness of dive sites within MPAs, leading to improved marine spatial planning for sea turtle conservation (*Wallace et al., 2010*; *Wallace et al., 2023*).

The two bycatch reports were submitted by the same fisher, approximately four years apart. According to the fisher, the loggerheads were found dead when they retrieved the catch using coastal bottom-set gillnets targeting finfish and stingrays. Most fishers in the BLP employ gillnets set for as long as three days leading to likely drowning if turtles become entangled (*Senko, Jenkins & Peckham, 2017*). The bycatch reports suggest the need for incentivized voluntary bycatch monitoring to establish a baseline understanding and co-design of effective mitigation strategies (*Morales-Bojórquez et al., 2021*). Most fishers do not report bycatch for fear of seasonal closures, loss of fishing permits, or fines (*Aguilar-González et al., 2012*; *Senko et al., 2011*; *Senko et al., 2014*; *Senko, Jenkins &*

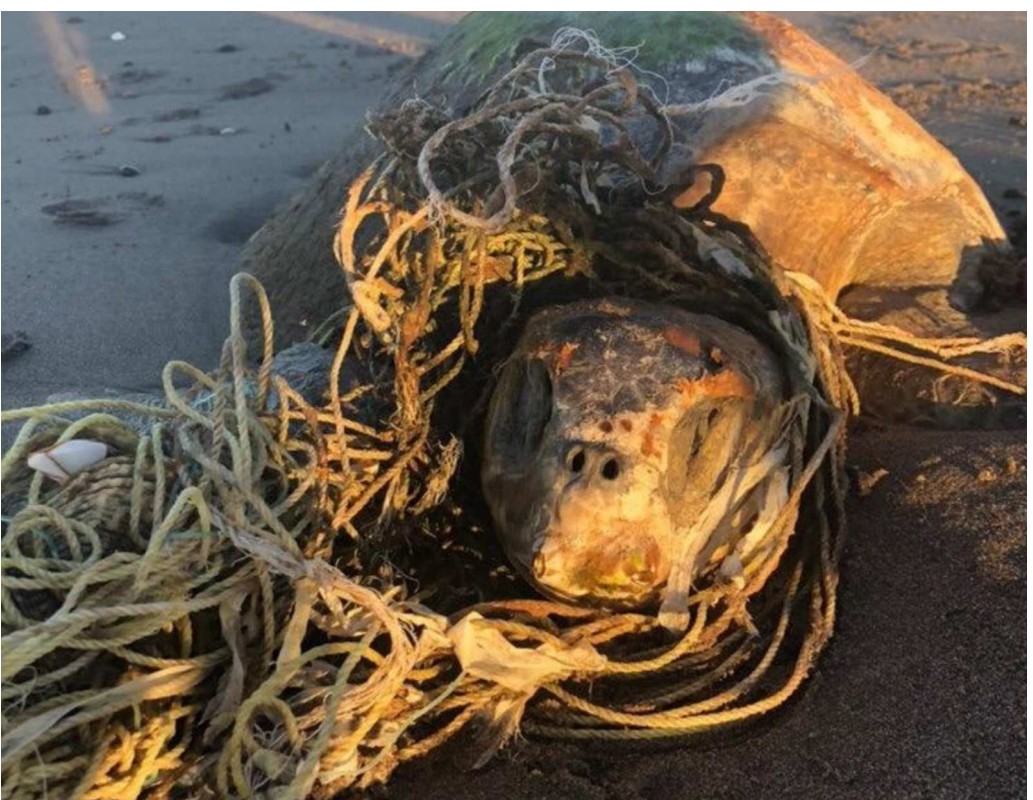

**Figure 6** **Stranded loggerhead turtle reported via social media with evidence of fishing net debris.** ID 08. Photo provided by Authors.

*Peckham, 2017*). Our results suggest that even a small number of reports from fishers can significantly enhance our understanding of species presence in fishing areas by contributing fisheries-dependent data through a citizen science approach (*Fulton et al., 2019*). In fact, this was observed by just one team of three fishers in Sinaloa who reported loggerheads as bycatch in the shark drift gillnet fishery, leading to new discoveries of foraging habitats in the GOC (*Zavala-Norzagaray et al., 2017*).

The six reports of dead stranded turtles were opportunistically received from various coastal residents who encountered the turtles on a beach. Given that most sightings involved stranded dead turtles, this likely reflects the greater ease of discovering a dead turtle washed ashore compared to spotting one underwater, particularly considering the pelagic nature of this species. The cause of mortality was not confirmed for any of the stranded turtles, although fishing nets and plastic were found on two individuals (Fig. 5). Considering the strandings occurred along the western side of the BLP, where several fishing camps are located and the two bycaught turtles were reported, it is plausible that these stranded turtles were discarded as bycatch and washed ashore. However, we could not determine the cause of mortality in any case, including the strandings our team verified in the field. Additionally, we cannot confirm that these turtles were utilizing the bay when they stranded; rather,

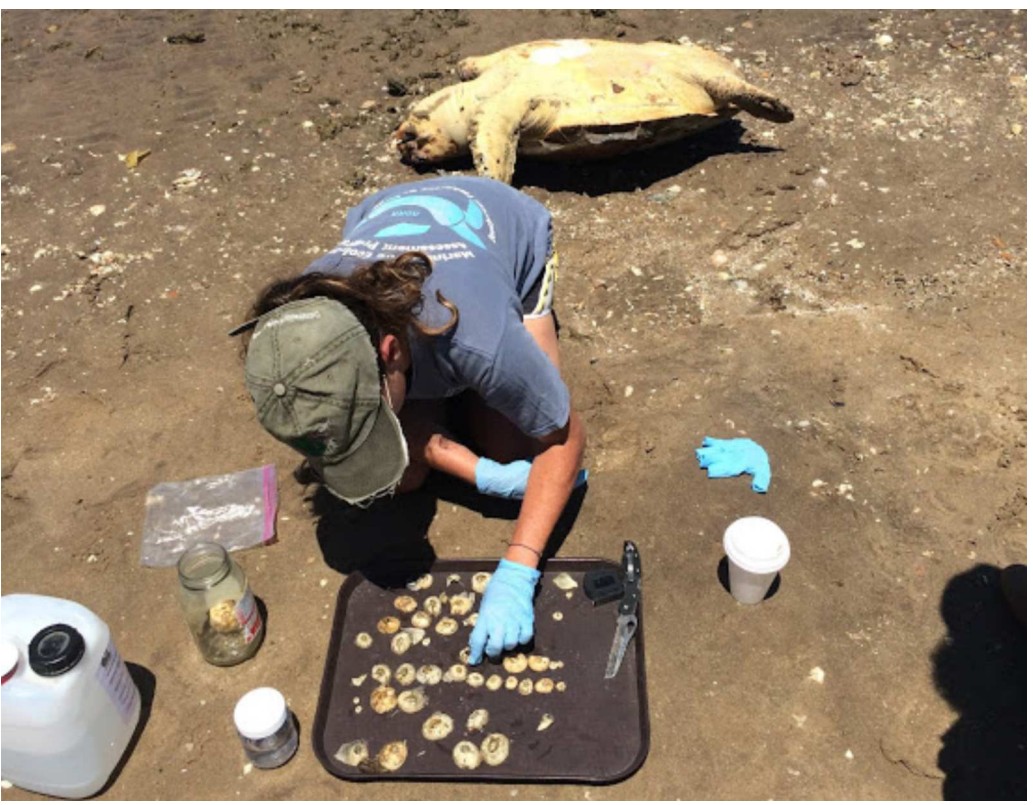

**Figure 7** **Stranded loggerhead turtle found in the Ensenada de LaPaz.** ID 01. Photo provided by Authors.

they could have been carried in by ocean currents from more pelagic areas, similar to what occurs in the Gulf of Ulloa (*Tomaszewicz et al., 2015*; *Mancini et al., 2012*).

The loggerhead sightings provided herein by both local and international coastal residents suggest a growing potential for community-based science initiatives in México. By leveraging data from multiple community groups, citizen science can enhance the accuracy and depth of ecological studies, leading to more effective conservation strategies. For example, in the Pelagie Archipelago of the Mediterranean, ecotourists and fishers reported sea turtle sightings and bycatch, which helped distinguish sea turtle occurrences within different areas of the archipelago (*Casale et al., 2020*). These initiatives demonstrate that engaging diverse community groups in scientific data collection can contribute to understanding and conserving marine species. By promoting community involvement, México can build on this model to enhance its own conservation efforts and ecological studies.

## ACKNOWLEDGEMENTS

We would like to acknowledge George Shillinger and Kristin Reed for their support of this project and management of the Sea Turtle Spotter initiative. We also thank Grupo Tortuguero de Las Californias, A.C. for their support and dedication to cultivating a
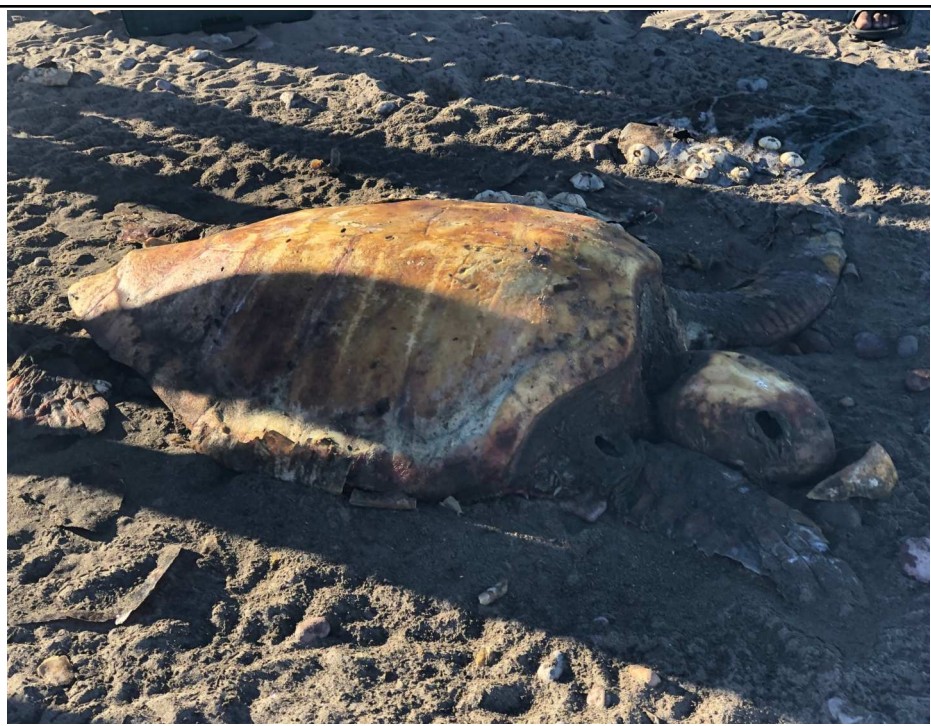

**Figure 8** **Stranded loggerhead turtle reported by coastal residents.** ID 05. Photo credit: Gabriela Flores Tom.

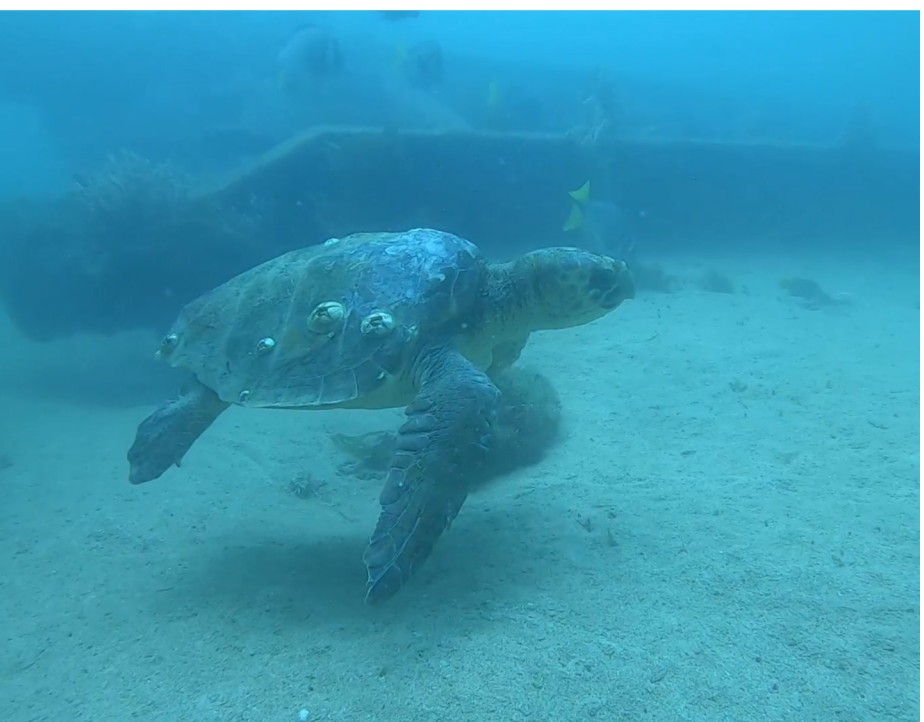

**Figure 9** **Live swimming loggerhead turtle** ID 09. Photo credit: Diana Martín Campilla.

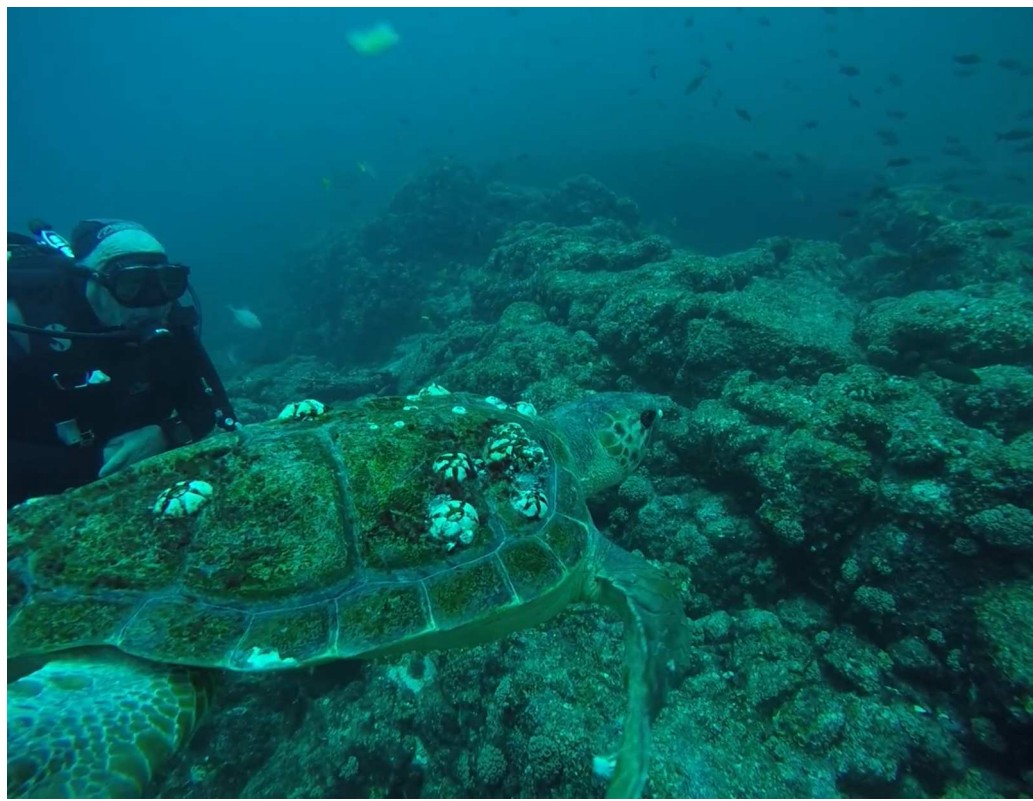

**Figure 10** **Live swimming loggerhead turtle ID 03.** Photo credit: Tamara J. Double.

collaborative network of sea turtle biologists, scientists, and fishers to inspire and support citizen science research. We also thank the SCUBA divers for donating the videos of loggerhead turtles, the coastal residents who reported the stranded turtles on the beaches, and the fisher for their trust to report the two loggerhead bycatch turtles to scientists. Finally, we thank three anonymous reviewers for their helpful comments and suggestions which improved the final manuscript.

### Funding

This work was financially supported either in-kind or monetary grants by Upwell Turtles, La Duna Ecology Center, and The Rufford Foundation. Upwell Turtles provided reimbursement for travel, workshop costs, and administrative services. The funders had no role in study design, data collection and analysis, decision to publish, or preparation of the manuscript.

### Grant Disclosures

The following grant information was disclosed by the authors:
Upwell Turtles, La Duna Ecology Center, and The Rufford Foundation.

## Competing Interests

The authors do not have any competing interests. Stephanie J. Rousso is a PhD candidate at CICIMAR-IPN from 2021 - current. Her research is funded by the Mexican government as a graduate scholarship. Stephanie is not currently, nor ever has been an employee of Upwell Turtles.

In 2019, Upwell secured an unrestricted fund, for 1 year, to support research initiatives in Mexico aimed at sea turtle conservation which Stephanie received and administered in Mexico to cover logistics, administration, and coordination of several research projects including Sea Turtle Spotter. Currently, Stephanie does not receive any funding, stipend, administrative support from Upwell.

Upwell supported the research outlined in this manuscript through administrative elements, such as graphics and administrative staff that were allocated to create the field guides of sea turtles used by citizen scientist, costs associated with hosting free public workshops, travel to confirm species identification in the field, visits to the fishing communities and other administrative support for research projects.

Agnese Mancini is employed full time by Grupo Tortuguero de Las Californias A.C.

## Author Contributions

- Stephanie J. Rousso conceived and designed the experiments, performed the experiments, prepared figures and/or tables, authored or reviewed drafts of the article, and approved the final draft.
- María Dinorah Herrero Perezrul analyzed the data, authored or reviewed drafts of the article, and approved the final draft.
- Agnese Mancini performed the experiments, analyzed the data, authored or reviewed drafts of the article, and approved the final draft.
- Alan A. Zavala-Norzagaray analyzed the data, authored or reviewed drafts of the article, and approved the final draft.
- Jesse F. Senko analyzed the data, authored or reviewed drafts of the article, and approved the final draft.

## Data Availability

The raw data are available in Table 1 (coordinates) and in Figs. 3, 4, 5, 6 (photos).

## Supplemental Information

Supplemental information for this article can be found online at http://dx.doi.org/10.7717/peerj.18203#supplemental-information.

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
