# Peer review of "Citizen science enhances understanding of sea turtle distribution in the Gulf of California"

_PeerJ, doi:10.7717/peerj.18203_

## Round 0.1 · original submission · Major Revisions

Dear Dr. Rousso,

After this first review round, I believe reviewer #1 raised serious potential flaws that may hinder your manuscript's acceptance to be published in PeerJ. On the other hand, despite the issues raised by reviewer #1, you have also obtained two "minor review" decisions. Considering all three decisions, I will grant you a major review. Still, please note that the acceptance of your manuscript is conditioned to the improvement of the text and the provision of compelling arguments to reviewer #1 regarding why your research should be published despite the raised flaws. If you can improve the text in the direction indicated by the reviewers and provide compelling arguments to reviewer #1 (if they accept the manuscript as valid after a second review), I will not object to accepting it for publication. Otherwise, in case reviewer #1 is not satisfied after the new review round, I am afraid I will have no other path other than rejecting your manuscript.

I really hope you are able to provide comelling arguments that convince reviewer #1 about the publication of this research.

Sincerely,
Daniel Silva

Reviewer 1 ·

Basic reporting

The manuscript is mostly well written and easy to understand. The English is professional and clear.


The article structure could be improved. The introduction does not have a logical flow. The first paragraph introduces Citizen Science, the second introduces Mexico (before a brief jump to California), the third returns to citizen science, and the fourth to Mexico. It would flow better to start with citizen science, then turtle cit sci, before moving in to the regional/national level.
The Methods second could be similarly re-ordered, starting with study site, then cit sci reporting methods, then validation methods.

In-text references are often incomplete (e.g. line 230 - Zavala et al. 2017, missing second surname)
or not included in reference list (line 63 - Alvarado et al. 2020)

Specific comments with line number.
#63 - Alvarado reference not in reference list
#75 - is citizen science "anecdotal"? Surely TEK or LEK is anecdotal. Citizen science has a structure that moves beyond anecdotes.
#85-87 - again, start global then go national with regulations and categories
#108 - What is BLA? Not referenced
#126 and others - I recommend using "fishers" instead of "fishermen" as a more inclusive term.
#131 - "Exchange" citation - I believe this should be Hoyt Peckham´s report?
#145 - "gTC"
#160 - "BLP" I guess is Bay of La Paz? But not defined until line 219
#168 - numbers one to nine written as words
#165 - "we confirmed as loggerheads" - who is "we"?
#176 - missing brackets on date
#189 - it is, not "it´s"
#230 - incomplete author name
#276 - incomplete author name
#289 - "1" should be "one"
#322-328 - in my PDF copy there is a strange formatting gap in the acknowledgement section.
References - follow journal style guide to ensure brackets and periods are used correctly.

Experimental design

I believe the paper´s methodology has design flaws that impact the validity of the conclusions.
While the paper makes an important contribution presenting a first record of loggerhead turtles in La Paz Bay, the bay is small and the turtle has already been documenting in neighbouring areas in the Gulf of California, the Pacific and southern Mexican Pacific as discussed on lines 108-112.

The same size is very small (10 turtles) to support the conclusion that there is great risk to turtles from the fishing camps or other sources. This does not mean I believe there is no risk to the turtles, but the results presented can not be extended to this conclusion. Stating that 80% (or eight) turtles were confirmed dead suggest survivorship bias (ie dead turtles are easier to be found/more likely to be reported) which the authors do not address in the methodology. The only swimming turtles were recorded by one diver who naturally would see swimming turtles in the MPAs he/she frequents. The dead turtles were reported by one conscientious fisher and (one?) resident on the beach. Are people more likely to report a dead turtle over a live one (probably)? I´m not sure whether eight dead turtle sightings over four years are sufficient to draw these conclusions and the authors make no discussion of the strengthens and weaknesses of the experimental design.

Citizen science is a valuable tool, and range extensions/first records are important contributions to the literature, but the authors should be careful with the conclusions drawn from a small sample size and possible recording biases.

Validity of the findings

The data provided is sound to support first records of loggerhead turtle, but limitations exist, as discussed above, on the conclusions drawn from the findings.

Additional comments

Citizen science remains a powerful tool for first records and range extensions. Publishing this information without drawing wider, possibly unsupported contributions could be a good contribution to the literature.

Reviewer 2 ·

Basic reporting

This is an interesting paper that highlights the value of citizen science in obtaining information on sea turtle distribution, specifically loggerheads, in the Gulf of California. The manuscript also examines the relevance of collaboration between scientists and various community groups to reduce knowledge gaps and develop better conservation strategies for sea turtles.
The authors did a good job of introducing the value of citizen science, but some sentences are not well-connected in certain sections. I also have concerns regarding the interpretation of the results, particularly in terms of the importance of the Bay of La Paz for loggerhead sea turtles. For example:
Lines 50-52: You mentioned that major organizations have published guidance documents that address citizen science, but they do not mention the purpose of these guidelines. What are they trying to accomplish?
Lines 52-59: It is clear that citizen science is relevant to the UN Sustainable Development Goals, but how does your work specifically contribute to SDG#14? Please explain.
Lines 63-64: While the source of this information is from 2020, the data itself is from 2014. Are there more recent data on citizen science in Mexico? CONABIO and other organizations such as COBI have been working extensively in promoting citizen science over the past decade.
Line 155: How many reports have you accumulated? I know you are focusing on loggerheads, but it is important to highlight the amount of data that citizen science can provide, not just for one marine turtle species.
Line 198: You are missing a sentence highlighting your main result, which is the presence of loggerheads in BLP.
Lines 202-203: How certain are you that these turtles died within the BLP and were not transported by ocean currents from elsewhere? This accounts for 60% of your turtles, and if there is a chance they came from somewhere else, the eligibility of BLP as an IMTA might not be as strong as you suggest.
Lines 223-227: While the idea of considering BLP as an IMTA is great, I do not think your current data on loggerheads is sufficient for its eligibility. However, you can recommend what studies are necessary to obtain enough data to prove that the IMTA criteria are met.
Lines 265-266: Can the stranded turtles also be an indicator of higher bycatch?
Lines 273-274: You have already made this statement in previous lines. I suggest rewriting this paragraph to make it clear that the bycatch and stranding data suggest a higher bycatch rate than previously expected.
Lines 278-279: What else do fishers need to participate in? Perhaps the application of good fishing practices, such as checking the nets constantly for entangled turtles and other bycatch species.
Line 287: Your stranded data cannot be used to infer spatial distribution; you are not sure where these turtles came from. Your data can only attest to the presence of loggerheads in BLP.
Other minor details on your figures and tables:
Line 478: It should say "Bay of La Paz, Baja California Sur, México…"
Line 481: Please add: "Fish icons represent fishing camps."
Line 484: It seems the following text is missing: "Fish icons represent fishing camps and colored diamonds represent citizen science reports." Also, should Figure 2.1 be listed as Figure 3?
Line 489: The word "turtle" should not be capitalized.
Figure 2: Although you mentioned in your text that BLP is defined based on the work of Del Monte Luna, it would help the readers to see these limits in your figure to better understand the size of the area. "Alfonso Basin" is misspelled on the map.
Figure 2.1 (3?): "Alfonso Basin" is also misspelled on the map.

Experimental design

No comment

Validity of the findings

No comment

Reviewer 3 ·

Basic reporting

The manuscript presents important information on the distribution of the endangered loggerhead sea turtle Caretta caretta which could lead to proper conservation programs.
Literature is appropriate.
The manuscript follows an appropriate structure.

Figure legends should be adjusted. No Figure 7 (this is probably Figure 6 and the text should be changed) and Figures 4, 5, and 6 had a and b.

Experimental design

Research question well defined, relevant, and meaningful

Validity of the findings

Conclusions are well stated

Additional comments

My major concern is the quantity of valid reports. I understand that the endangered loggerhead sea turtle is an endangered species and hard to spot on the ocean, but for a citizen project, an average of 2 sightings per year is very low. Maybe in the materials and methods mention the total reports that were cleaned in order to get 10 valid reports.
Also, I noticed the lack of reports for 2020, I assume this is due to the lockdowns of Covid, maybe mention that.

---

## Round 0.2 · accepted · Accept

Dear Dr. Rousso,

After this new round of reviews, all reviewers believe the manuscript has improved. Therefore, I am pleased to recommend the publication of your manuscript in PeerJ.

Sincerely,
Daniel Silva

Reviewer 1 ·

Basic reporting

The authors have made significant improvements with this second version of the paper. The text is clearer, easier to read and follows a more logical order.

Experimental design

As of previous comments, the methodology in of itself is sufficient and clearly described. In this version the authors have refrained from extrapolating beyond what is attainable from their methodology.

Validity of the findings

Greatly improved results and discussion. No improvement suggested.

Reviewer 2 ·

Basic reporting

In this second version of the manuscript, the authors have done an excellent job of incorporating the reviewers' suggestions and addressing concerns regarding the interpretation of results derived from a small number of records. They clearly articulate the value of citizen science in gathering information on sea turtle distribution and in providing the first recorded sighting of loggerhead turtles (Caretta caretta) in the Bay of La Paz. The narrative is now clear, well-structured, and supported by sufficient references to strengthen their discussion and conclusions. I do not have additional comments.

Experimental design

The authors succinctly and clearly describe the methods used to validate the sighting reports and classify them based on the source of information, the status of the organism, and the type of report.

Validity of the findings

I find their results and assumptions to be well-founded and supported by both the previous and newly included references, following the reviewers' recommendations. Although the scope of the original paper has shifted based on these recommendations, it still holds unique value as it provides the first record of loggerheads in the Bay of La Paz and underscores the significant contribution of citizen science in achieving this.

Reviewer 3 ·

Basic reporting

The authors addressed all the comments and suggestions made by the reviewers.

Experimental design

N/A

Validity of the findings

N/A

Additional comments

N/A